# Application of Internet of Things (IoT) for Optimized Greenhouse Environments

**Chrysanthos Maraveas** (ID) **and Thomas Bartzanas** *(ID)

Farm Structures Laboratory, Department of Natural Resources and Agricultural Engineering, Agricultural University of Athens, 11855 Athens, Greece; maraveas@aua.gr
* Correspondence: t.bartzanas@aua.gr

**Abstract:** This review presents the state-of-the-art research on IoT systems for optimized greenhouse environments. The data were analyzed using descriptive and statistical methods to infer relationships between the Internet of Things (IoT), emerging technologies, precision agriculture, agriculture 4.0, and improvements in commercial farming. The discussion is situated in the broader context of IoT in mitigating the adverse effects of climate change and global warming in agriculture through the optimization of critical parameters such as temperature and humidity, intelligent data acquisition, rule-based control, and resolving the barriers to the commercial adoption of IoT systems in agriculture. The recent unexpected and severe weather events have contributed to low agricultural yields and losses; this is a challenge that can be resolved through technology-mediated precision agriculture. Advances in technology have over time contributed to the development of sensors for frost prevention, remote crop monitoring, fire hazard prevention, precise control of nutrients in soilless greenhouse cultivation, power autonomy through the use of solar energy, and intelligent feeding, shading, and lighting control to improve yields and reduce operational costs. However, particular challenges abound, including the limited uptake of smart technologies in commercial agriculture, price, and accuracy of the sensors. The barriers and challenges should help guide future Research & Development projects and commercial applications.

**Keywords:** Internet of Things; technology; greenhouses; agriculture



## 1. Introduction

This review article synthesizes current scholarly research on the application of IoT for optimized greenhouse environments with an emphasis on IoT-mediated and optimized microclimates for crop production. The focus on IoT is grounded on the immense contribution of technology to modern civilizations after computers and the internet [1]. The immense contribution of IoT in agriculture and commercial greenhouses could be linked to the integration of intelligent machines, actuators, sensors, unmanned aerial systems, radio frequency identification (RFID) devices, big data analytics, artificial intelligence, and satellites [2], and this has facilitated its widespread application in various agricultural and non-agricultural applications, including intelligent farming and frost prevention in greenhouses [3], intelligent control of greenhouse structures [4], fire hazard prevention [5], transition to agriculture 4.0 [6], precise-control of nutrients in soilless greenhouse cultivation [7], smart cities [8,9], emission monitoring [10], distributed/decentralized energy storage, solar-powered sensors [11], smart feeding, shading and lighting control, and security [12–14]. The widespread adoption of IoT in smart greenhouses and precision agriculture has been partly augmented by the development of highly efficient communication protocols such as MQTT Protocol (Message Queuing Telemetry Transport), which have gradually phased out HTTP (Hypertext Transfer Protocol) [15]. MQTT is capable of running on lower bandwidth, which translates to lower overhead protocols.

Despite the widespread adoption of IoT systems in smart greenhouses, there has been an inadequate understanding of how the technology can optimize greenhouse environ-

ments, especially in tropical regions that experience severe temperature fluctuations. Most of the research and development has been localized in developed nations, with direct access to IoT systems and resources. In addition to the gaps in the understanding, the contribution of IoT systems in smart greenhouses in tropical regions is inadequate.

*Emerging Smart Technologies and Their Domino Effects on Precision Agriculture*

There are multiple emerging technologies that are predicted to significantly impact the future of precision agriculture. The technologies that would have the most notable impact include cloud computing, edge computing, fog computing, embedded software, embedded systems, cyber-physical systems (CPS), Wireless Sensor Networks (WSN), Big Data Gateway, Machine to Machine (M2M), Human to Machine (H2M), LoRa Protocol (LoRaWAN), ZigBee/Z-Wave Radio Frequency Identification (RFID), Gateway General Packet Radio Service (GPRS), Application Programming Interface (API), Advanced Encryption Standard (AES), and Digital Twins. The intersection of the different ICTs, cloud computing, WSN, geo-location, satellites, and computer-to-human interfaces is illustrated in Figure 1. The potential role of these technologies in water and energy conservation and the long-term cost-benefits in greenhouses are briefly considered.

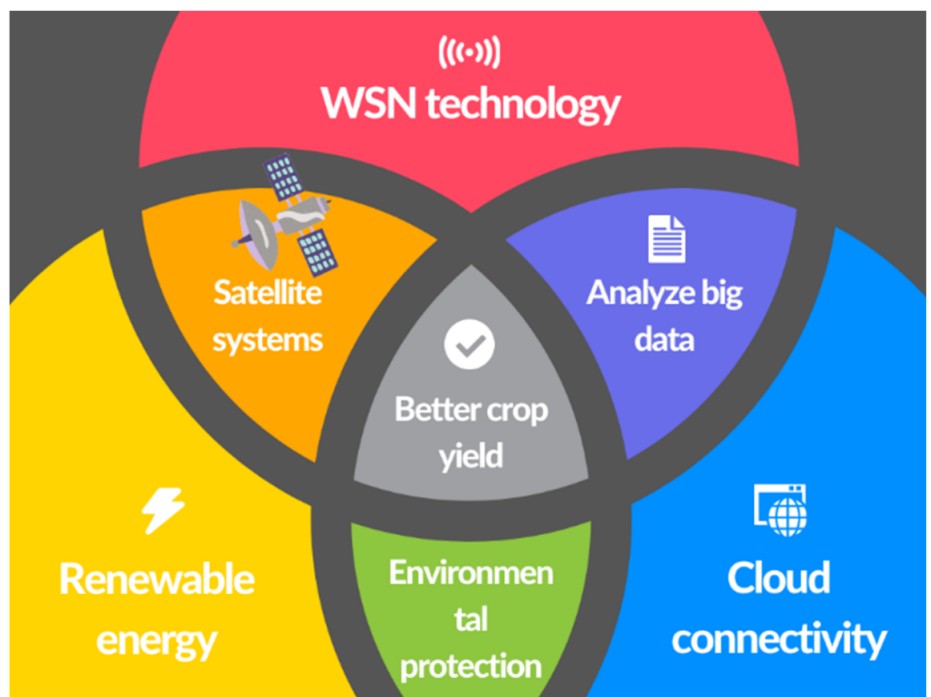

**Figure 1.** The intersection of cloud computing, WSN, geo-location, satellites, and computer-to-human interfaces.

Cloud computing has been proven useful in smart greenhouses and precision agriculture [16–19]. The storage of sensor data in the cloud and the integration with smart technologies for the remote monitoring of plant water levels, nutritional content, soil pH, humidity, and temperature, has translated to significant cost savings and improvements in yields [18]. The pathways to better efficiency are multidimensional. For example, the farmers can access historical predictive analytics data from institutions [20] and use the data to establish supply and demand trends across different product markets. The predictive data also provides farmers with real-time information about weather conditions, thus mitigating the adverse effects of climate change and global warming. However, there are critical drawbacks associated with the extended use of cloud computing in areas with insufficient network coverage and low internet speeds.

Frequent upload and download of data result in the slow transmission of the data. The challenge can be partly offset through the incorporation of edge computing. Zhang, Cao, and Dong [21] confirmed that edge computing could offset the strain on the network infrastructure by sharing the load of the cloud server; this, in turn, translated to lesser delay. The elimination of the technology constraints increases the potential areas of application of IoT, especially in the intelligent management of crops and agricultural machinery, the safety traceability of agricultural products, and pest identification. The positive assessment of the role of edge computing by Zhang, Cao, and Dong [21] is consistent with Akhtar et al. [22] and O'Grady, Langton, and O'Hare [23]. Both studies concurred that edge computing would have a positive impact on the agricultural industry. In contrast to cloud computing in agriculture, which is well-grounded, edge computing is an emerging application; edge computing is a nascent field. In 2021, the first prototypes of edge computing were still under development. In addition, there has been no reliable and widespread validation of edge-driven services in farms. Even if the technologies were available, new challenges would emerge, especially in the standardization of IoT platforms [23]. The newness of edge computing coupled with the drawbacks of cloud computing clearly illustrates that the future of IoT in agriculture depends on global technological advances and investments in research and development.

## 2. IoT and the Mitigation of the Adverse Effects of Climate Change in Agriculture

Traditional agriculture is characterized by unnecessary human interaction resulting in higher labor costs and susceptibility to severe weather events due to limited integration of data-driven decision support systems. IoT offers a reprieve through AI and machine learning-based energy- and water-saving measures, automated farm operations, and mechanization to resolve crop monitoring challenges [24,25]. The role of technology in future farming cannot be negated in light of the adverse effects of global warming and climate change on global food production and food safety. Unusual climate events such as extreme temperatures and precipitation have been associated with a significant reduction in agricultural production in the US and other nations.

In 2012, cherry farmers in the state of Michigan lost an estimated 220 million USD due to severe weather changes [26]. The climate-related agricultural losses are anticipated to persist over the long term, considering that pests, fungi, and bacteria thrive best under hot weather conditions. Conservative estimates indicate that farmers would spend 11 billion USD more to fight pests and diseases yearly due to climate change [26]. Any increase in the cost of production would have a higher adverse effect on agrarian communities in developing nations [27]. The impact of climate-related factors on agriculture would not be confined to farmers, given that the additional costs would be passed on to consumers. The problem was not unique to the US or advanced economies in the western hemisphere. Kava et al. observed similar challenges noted in Greece, where the arable land area had decreased [28]. Climate change-related disruption in weather patterns coupled with sociodemographic factors had contributed to severe food shortages and malnourishment [27]. The diversity of the challenges across developed and emerging nations underscores the need for innovative and technological solutions to mitigate the climate change-induced disruptions in agriculture.

Multiple options have been explored in the recent past, including greenhouse-based farming. As of 2021, there were 60,000 acres under greenhouses in Greece, and the distribution was concentrated in areas that were significantly impacted by climate change and global warming, such as Messara, Ierapetra, and Crete [28,29]. Similar models were adopted by Gulf nations with limited arable land. As of 2016, 3019 ha were under greenhouses in Saudi Arabia, yielding about 252,824 ton of fruit-bearing vegetables [7]. The Intergovernmental Panel on Climate Change forecasts that the prices for agricultural production will increase by 1–29% by the year 2050 [30]. The challenges associated with

climate change in agriculture could be partly offset by the optimization of greenhouse environments using IoT; this means that advanced technologies would be indispensable to global food supply chains [28].

The potential IoT-based solutions for modern farming challenges would be centered around the four critical areas of the application of IoT systems in smart agriculture and greenhouses, namely the maintenance of an ideal microclimate for ideal plant growth, enhanced irrigation and fertilization practices, control of infection, and improved security [31]. Security in agricultural production could be achieved using infra-red cameras, unmanned aerial vehicles/systems for remote monitoring [2], optical monitors, infra-red, and thermal sensors placed strategically across large farms to mitigate crop losses due to invasion by wildlife (such as birds, goats, and buffaloes) [32,33]. The use of intelligent security solutions is augmented by cost considerations—sensors for remote monitoring coupled with robotics and big data are affordable and efficient compared to human labor [34]. The commercially available sensors have demonstrated that it was feasible to reduce costs and improve efficiency through technology.

A key goal moving forward would be to minimize manual interventions, improve yields, and optimize the usage of resources and agrochemicals [35,36]. However, the actualization of the potential benefits associated with sensors would be contingent on the optimization of specific parameters and the integration of accurate sensors for monitoring water and moisture content and plant physiology [35]. Preliminary findings show that IoT systems offer great promise in precision agriculture. For example, the weather data-driven decision support systems alerted farmers when it was most appropriate to apply fungicides [37]. The timely application of fungicides helped to mitigate the risk of late blight, and this translated to direct cost savings of about 500 USD/acre [37]. Similar benefits were documented with the adoption of weather data-driven decision support systems coupled with electrical capacitance sensors for soil–water balance and soil–water content. The design translated to 25% cost savings on irrigation for wheat farmers. The technology was also proven helpful in the application of fertilizers using optical sensors that analyzed the plant chlorophyll content (a predictor of plant nitrogen levels). The plants with more significant nitrogen deficiency received higher quantities of nitrogen-rich fertilizer [37]; this resulted in the optimal utilization of fertilizers and greater yields.

Khudoyberdiev et al. and Miller and Cappuccio's [35–38] assessment of the cost–benefits of sensors in greenhouses are in line with Agrawal and Kate [39], who affirmed the link between sensors and product yields. On average, greenhouse-based agricultural production improves yields by 10–12% [5]; this is supported by empirical evidence—greenhouses in Saudi Arabia yielded 252,824 tons of vegetables and fruits in 2016 [7]. Current forms of IoT systems for precision agriculture were less adapted to semi-arid regions [40]; this means that the functionality of sensors in desert areas remains an issue of concern. On the downside, there is little evidence of technology adoption and precision agriculture in developing nations [41]. The poor adoption of emerging innovations in developing countries helps to explain the consistent low yields, contributing to the gaps in food security between developed and emerging nations. The commendable progress documented in Greece, Saudi Arabia [7], and other countries despite the weather/climate-related technological limitations validates the need for smart and intelligent agriculture to improve crop yields in emerging nations and developed countries with extreme temperature variations.

The case for customized solutions and interventions was augmented by the potential improvements in yields. The standard improvements in yields range between 10 and 12% [5], but better performance can be achieved through the optimization of plant growth factors, technological advancements in the reliability of the sensors, and cost control. For example, farms with scattered greenhouses might be obliged to adopt a scattered rather than intense deployment of sensors due to cost considerations [42]. The concerns made

concerning the cost of IoT in agriculture were in line with Zamora-Izquierdo et al.'s [40] research on the link between precision agriculture and cost. In most cases, IoT in agriculture was expensive and out of reach for the smallholders—the backbone of the global agricultural system [43].

The inability of smallholder farmers to invest in IoT technologies could be attributed to the prevailing uncertainty about the selling prices of agricultural produce due to variations in the market prices, lack of defined parameters about energy and water allocations, and energy use [24]. The tight production margins provide a minimal incentive for smallholder farmers to invest in new technologies such as IoT; this contrasts with large commercial producers who can easily acquire IoT systems. The comparative analysis of the two clearly indicates that cost was a critical impediment to the widespread use of IoT infrastructure [24]. The challenges identified by Villa-Henriksen et al. [24] were corroborated by Madushanki et al. [25]. In the latter case, it was postulated that the benefits outweighed the risks given the potential of IoT infrastructure in catalyzing smart farming and urban greening.

Under an intense deployment of sensors, a greenhouse/farm is fitted with multiple interconnected sensor nodes across different points, which facilitate real-time monitoring of the areas of interest using WSNs [42]. In contrast, a scattered configuration is characterized by the less-intense distribution of sensors to save costs. However, the cost savings involve a tradeoff with accuracy and autonomous IoT system response; this means that key data might not be captured. Sharma et al.'s [42] outlook concerning the accuracy and autonomous IoT-system response epitomizes the concerns raised by different stakeholders about precision agriculture. Attempts to address these issues included autoregressive integrated moving average (ARIMA) models for predicting and forecasting anomalies related to sensor failure [44]. Alternatively, aging or faulty sensors could be repaired semi-automatically using SensorTalk innovation that integrates AgriTalk graphical user interface (GUI) and temperature calibration cyber devices [44]. The autonomous and remote repair of sensors is a viable solution for smallholder farmers and large commercial farms.

## 3. Current Research on IoT in Greenhouses

### 3.1. Optimization of IoT Systems for Smart Greenhouses and Intelligent Farming

There is broad consensus among scholars that the optimization of IoT systems for greenhouses requires the careful selection of sensors, data acquisition, optimization, determination of the desired settings, and rule-based control (see Figure 2) [35,36,45]. Considering that the conditions have to be optimized for different horticultural crops, there is a need for advanced techniques for data acquisition using Kalman filter prediction and related techniques that can forecast future environmental conditions using historical data. In reality, the regulation of each parameter (temperature, moisture, pH, pesticides, humidity, UV radiation, rain, $CO_2$, and pressure) in smart greenhouses is a challenge [38,46], especially in cases where there is inadequate historical data.

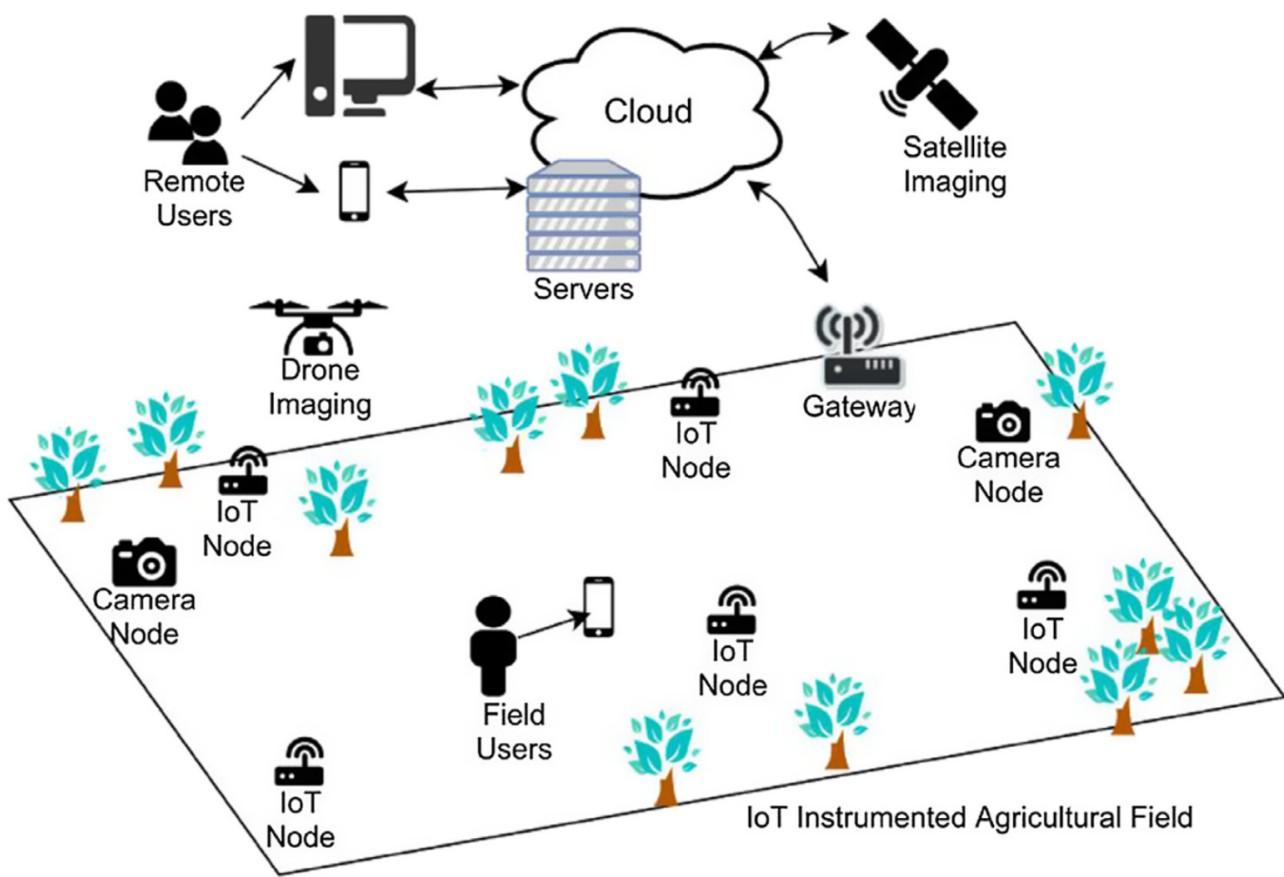

**Figure 2.** Core aspects of IoT-based systems for smart greenhouses [45] (Reproduced with permission from publisher (Elsevier) from Popović et al. (2017)).

### 3.2. IoT-Based Sensors for Smart Greenhouses

Presently, there is a wide array of IoT-based sensors for smart greenhouses, including plant growth sensors, temperature and humidity sensors, insect detection sensors, soil temperature, pH, and moisture sensors, and solar radiation, atmospheric pressure, wind speed, and $CO_2$ (and other gas) sensors, which rely on Bragg, piezoelectric, electrochemical, electromagnetic [31], and fiber-optic technologies for accurate assessment of the desired parameters [47] (see Table 1). The parameters of interest include different wavelengths of light, photocurrent, fluorescence intensity, the fluorescent signal emitted by plant chlorophyll, optical density, and the electrochemical signal generated by enzyme-catalyzed redox reaction (SHA principle) [1]. Advances in research and design have resulted in the development of electromagnetic sensors for analyzing chlorophyll values and nitrogen concentration in plants; this approach relies on light reflectance and pulsating laser diodes [31]. The technique was also proven useful in measuring the real-time plant physiology, including a plant's vegetative index, nutritional requirements, electrical conductivity, and magnetic susceptibility and conductivity (quad-phase) [31].

**Table 1.** Common types of sensors used in smart greenhouses [30].

| Application of Sensors | | Sensor Models and Manufacturers |
|---|---|---|
| Crop monitoring | Growth | Cyber-shot DSC-QX100 (Sony Electronics Inc., Tokyo, Japan), Parrot Sequoia (MicaSense Inc., Seattle, WA, USA) |
| | Insects and disease detection | FLIR Blackfly 23S6C (FLIR Systems, Wilsonville, OR, USA) |
| | Active canopy sensor | ACS-430, ACS-470 (Holland Scientific, Inc., Lincoln, NE, USA) |
| Substrate monitoring | Soil temperature, soil moisture | DS18B20 (Maxim Integrated, San Jose, CA, USA), VH400 (Vegetronix, Salt Lake City, UT, USA), HL-69, ECH2O-10HS (METER Group, Pullman, WA, USA) |
| | PH | E-201 (Shanghai REX Sensor Technology Co, Shanghai, China) |
| | Chemical elements (e.g., nitrate, nitrogen, etc.) | SEN0244 (DFROBOTS, Shangai, China) |
| Environment monitoring | Air temperature, air humidity Solar radiation | DHT11, DHT22 (AM2302, Aosong Electronics Co., Ltd., Guangzhou, China) SQ-110 (Apogee Instruments, Inc., Logan, UT, USA) |
| | Rain | YF-S402 (Graylogix, Bangalore, Karnataka, India), YL-83 (Vaisala Corp., Helsinki, Finland) SE-WS700D (Lufft Inc., Berlin, Germany) |
| | Luminosity | BH1750 (Rohm Semiconductor, Kyoto, Japan), TSL2561 (Adafruit Industries, New York City, NY, USA) |
| | Atmospheric pressure | MPL3115A2 (NXP Semiconductors, Eindhoven, The Netherlands) |
| | Wind speed and direction | WS-3000 (Ambient Weather, Chandler, AZ, USA), (SparkFun Electronics, Niwot, CO, USA) |
| | $CO_2$ concentration | MG-811 (Zhengzhou Winsen Electronics Technology Co., Ltd., Zhengzhou, China), MQ135 (Waveshare Electronics, Shenzhen, China) |
| Other | Tracking | Mifare Ultralight NFC tag (NXP Semiconductors, Eindhoven, The Netherlands), Blueberry RFID reader (Tertium Technology, Bangalore, Karnataka, India) |
| | Localization | UM220-III (Unicore Communication Inc., Beijing, China) |

The regulation and monitoring of ambient conditions in a greenhouse are critical considering that excess heat or humidity translates to large-scale plant damage. The ideal plant ambient temperature and relative humidity for optimal plant growth were 35 °C and 95%, respectively [39]. Excess temperature and humidity are detrimental to optimal plant growth, given that they impair pollination, photosynthesis, leaf growth, and yield [41]. The optimization of these parameters in smart greenhouses is partly impaired by the limited accuracy of available sensors. The accuracy of selected sensors is limited to 2–25% [48]. Such low levels of accuracy could have disastrous consequences in the regulation of greenhouse microclimates, especially frost mitigation [3,49]. In other cases, the accuracy of the IoT sensors is 99% [4]. High levels of accuracy were achieved with smart stick sensors that provide live monitoring information about soil moisture and temperature. The information drawn from the smart stick sensors is synchronized with smart devices to obtain live feeds about variations in the physical parameters. The observations made by Rayhana et al. [4] concerning the high accuracy of smart sensors for greenhouses were aligned with Castañeda-Miranda et al. [49]. In the latter case, the accuracy of the sensors for frost suppression was linked to the reliability of fuzzy associative memory (FAM) and Artificial Neural Networks (ANN) [49]. The internal temperature in a greenhouse was measured using ANN, while the cropland temperatures were analyzed using fuzzy control, and this resulted in the activation of the water pump. The experimental data confirmed that the IoT system for frost mitigation based on ANN and FAM was >90% accurate relative to the Fourier statistical analysis of hourly data [49]. The reliability of the data was corroborated by inferential statistical measures.

New research has demonstrated that it was practical to sustain the high levels of accuracy reported by Rayhana et al. [4] through the optimization of the sensor resolution (minuscule changes in the input signal that can be detected) and precision (reproducibility of the same output with similar input) using automated calibration techniques such as sensor failure detection, identification, and accommodation (FDIA) in SensorTalk technology [32]. Considering that Rayhana et al. [4] and Castañeda-Miranda et al. [49] observed

that IoT sensors achieved superior performance in the optimization of the greenhouse microclimate, the low accuracy reported by Ryder et al. [48] could be attributed to the sensor properties or poor control of external variables. Such drawbacks can be offset by system maintenance and calibration with spectrophotometers [1] and customization of the system to match the local environmental conditions. Alternatively, the ongoing deployment of the 5G network would catalyze the expansion of IoT infrastructure and the deployment of advanced sensors with better precision [37]. From another perspective, the contradicting evidence on the accuracy of the sensors documented by Ryder et al. [48] and Wang et al. [1] raises fundamental questions concerning the reliability of sensors for agricultural applications, which might, in turn, affect the commercial uptake of sensors. The central role of accurate sensor data on IoT platform data integration, real-time visualization, and application prototyping is illustrated in Figure 3.

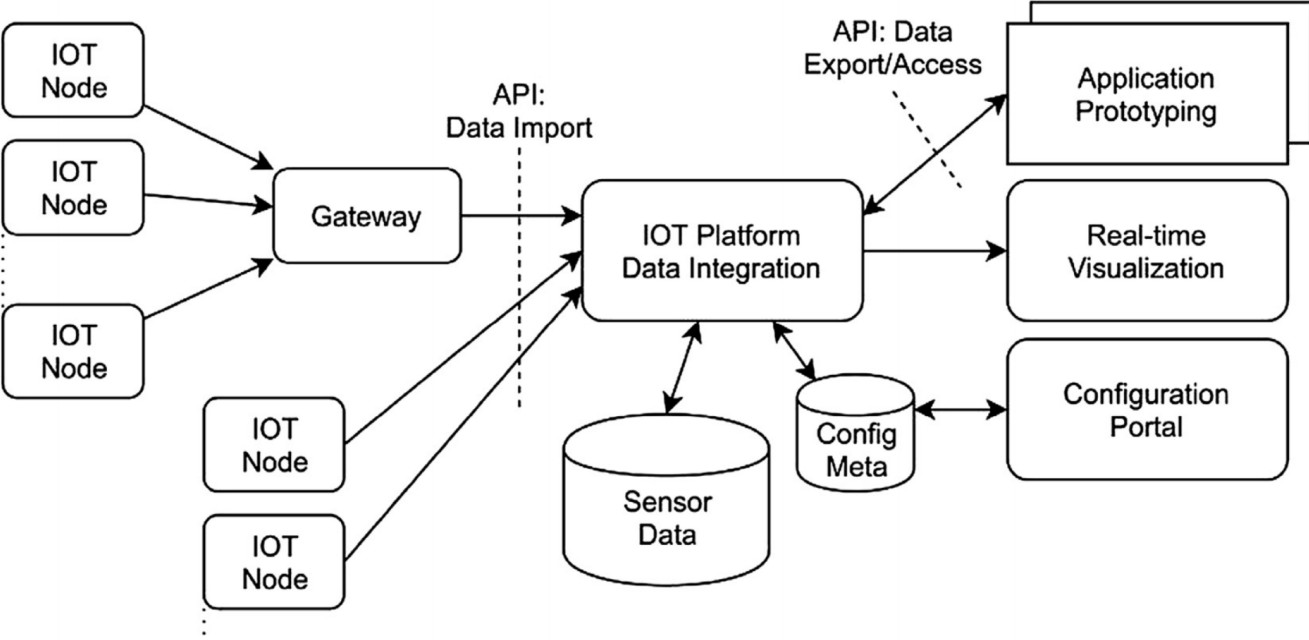

**Figure 3.** Sensor data and IoT platform data integration [45] (Reproduced with permission from publisher (Elsevier) from Popović et al. (2017)).

Following the critique of various arguments made concerning the reliability of sensors [50–53], the researcher notes that the accuracy of IoT sensors in agricultural settings was influenced by application- and context-specific factors. The worldview is corroborated by Wang et al. [1], who reported high accuracy in sensors designed to measure biological parameters in microalgae farms. The high accuracy of the sensors was linked to optimal design configurations [1]. For example, the potential adverse effect associated with outdoor temperature and other variables was offset by the integration of temperature control devices, including a multi-wavelength laser diode-photodiode for the estimation of the cell concentration [1]. The latter evidence underscores the need to choose the correct types of sensors and optimize performance through the integration of multi-wavelength laser diode-photodiodes, rRNA biosensors, microfluidic chips featuring nanocavity-enhanced photoelectrochemistry (Cu nanoparticles and the Cu-electrode), and multiprobe chips [1]. The need to optimize the performance of the sensors is validated by the cost implications associated with the errors. For example, sensor errors could result in higher energy usage in the regulation of the greenhouse microclimate [54]. In rare cases, the sensor errors might have a beneficial impact on the crop; poor assessment of the crop nutritional content could translate to the excess application of fertilizers and micronutrients, and this would automatically translate to better crop yields. The unexpected benefits associated with sensor

errors were demonstrated in a study of agricultural sensors in Dutch greenhouses [54]. In reality, it is imperative to appreciate the fact that such benefits are often isolated.

Despite the commercial availability of multiple sensors, the selection process is guided by certain considerations such as the cost, accuracy, power requirements, material properties, functionality under extreme conditions (excess moisture and precipitation), integration with different components, and technologies such as neural networks and machine learning [31,55–67]. The performance of different classes of sensors under extreme conditions has been augmented through the development of polymer membranes to protect the sensor components from particulate contaminants, dust, and water, which might compromise the accuracy and response time [48]. Proof of concept designs has been demonstrated for solar-powered sensors for smart greenhouses. Sahraei et al. noted that such sensors were suitable for remote areas without a connection to the grid [35]. Beyond the satisfaction of the inherent power requirements, PV-powered sensors contribute to the power autonomy and miniaturization of smart devices for greenhouses, which in turn translate to better cost savings. Even though the power needs have been resolved through the adoption of autonomous PV sensors, as noted by Sahraei et al., other challenges abound.

Emerging materials such as graphene have proven to be ideal materials for sensors for agricultural applications based on their high tensile strength, light weight, flexibility, and ecological friendliness [58]. However, the widespread use is inhibited by the absence of scalable methods for the commercial synthesis of the materials [58,59]. The reservations towards novel materials are further reinforced by empirical facts and market data. Despite the commendable progress made by start-up companies in graphene R&D, there have been significant inconsistencies in the properties of the materials of graphene produced on an industrial scale versus laboratory-scale graphene [60]. The inability to replicate and reproduce quality materials remains a critical impediment beyond the limits of emerging materials and technologies. There are varied drawbacks that limit the optimal placement of sensor notes for better coverage and communication among sensors [42]. The listed challenges have practical consequences on greenhouse sensors for soil and water analysis. Even though technological progress offers promising solutions, the accuracy of the sensors remains a challenge considering that the accuracy of the sensors is affected by stray signals associated with pressure, light, and humidity [48]. Even though layer optimization offers a practical solution, it has not been explored in detail.

### 3.3. Barriers to the Commercialization of IoT Technologies for Greenhouses: Internet Connectivity, Cost and Technological Limitations

The development of different classes of sensors by Unicore Communication, Ambient Weather, Chandler, NXP Semiconductors, and Apogee Instruments, among other manufacturers [38], provides a pathway to the commercialization of IoT systems for greenhouses. However, key barriers to technology acceptance must be resolved. Cisco and the International Telecommunication Union (ITU)'s report categorized the barriers into policy and technical domains (see Figure 4) [61]. The intersection of the two domains introduced a third dimension specific to the spectrum and bandwidth constraints, privacy, security, interoperability, and standards. For example, 3G and 4G internet technology are governed by UMTS and CDMA 2000 standards. In contrast, IEEE 802.11 and IEEE 802.15.4 regulate the operationalization of WiFi and LR-WPAN networks, respectively. Considering that these networks have different frequencies, data rates, and power requirements [24], it becomes challenging to standardize IoT infrastructure in the agricultural sector.

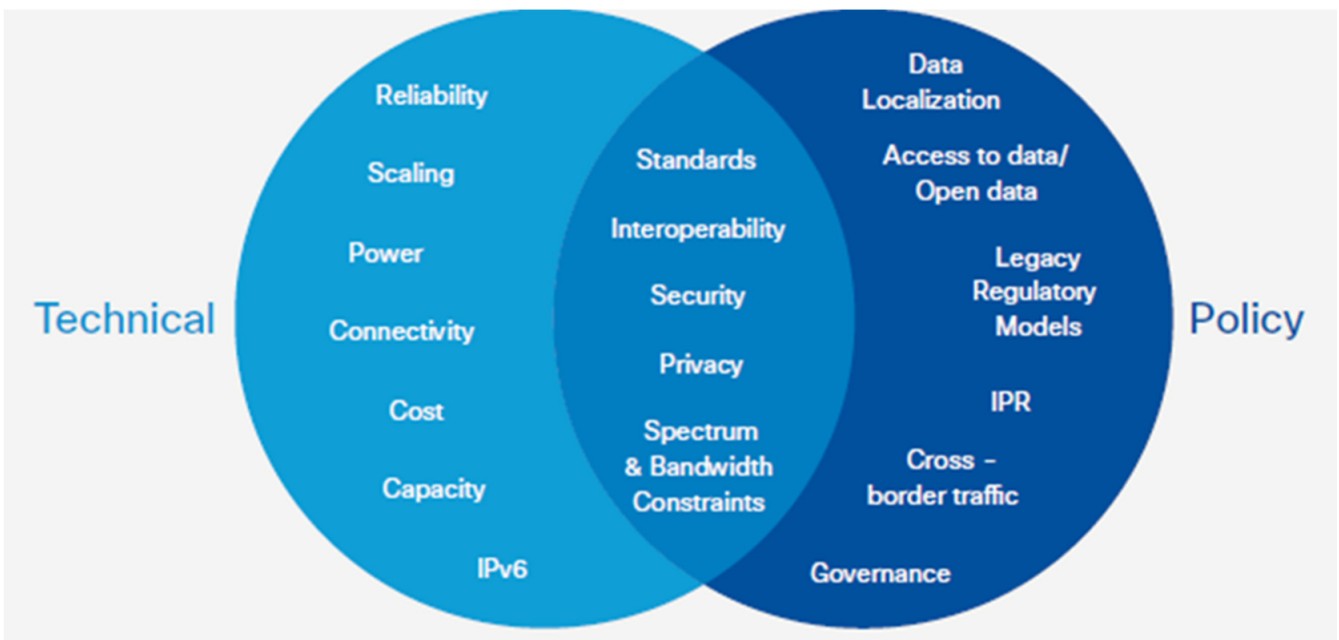

**Figure 4.** Policy and technological barriers to the global adoption of IoT systems in agriculture and beyond [53] (Reproduced with permission from publisher (Elsevier) from Molinara et al. (2021)).

As noted by Cisco and the International Telecommunication Union (ITU) [61], the reliability of IoT systems was a critical factor. In greenhouse farming, reliability was predicted by the accuracy and reliability of sensors in assessing the plant chlorophyll content and the cost-effective regulation of soil properties and improving the yield potential [1,40,44]. At present, IoT systems for agriculture rely on technologies available in other industries such as GPS positioning for the navigation of unmanned aerial systems, reflectance spectroscopy, and microwave sensing for agricultural sensors, but IoT infrastructures are unevenly distributed in developed and emerging nations [62]; this introduces new constraints in terms of spectrum, bandwidth, interoperability, and standards [61]. Since advanced economies had better mechanisms for overcoming the barriers to technology acceptance, the commercialization of IoT systems in greenhouses was feasible. On the downside, IoT deployment in developing nations and rural areas remains a challenge due to the cost-benefits associated with the commercialization of IoT infrastructures, as noted by Ruan et al. [65]. The latter observations were supported by industry and internet penetration data [64]. Since the start of the internet revolution in the 1990s and early 2000s, it has taken the global community an extended period to build stable internet infrastructures. However, by Q4 of 2021, only 56% of the global population had access to the internet [63]; this means that half of the global population was unable to unlock the IoT benefits in precision agriculture. The challenge might be resolved in the long-term using a low-earth orbit (LEO) constellation, which provides the potential for global-internet connectivity with diminished latency [64]. OneWeb and Starlink companies are on course to provide high-speed, low-latency broadband internet with low latency that is ideal for rural and remote areas [65,66]. However, LEO deployment has lagged, and there are varied concerns about global adoption.

The technological challenges had a more profound impact on smallholder farmers and big agricultural corporations in rural areas and nations with poor rates (<25%) of internet connectivity such as Afghanistan, Nepal, Pakistan, Central African Republic stands, and Turkmenistan [63]. The poor development of key network infrastructure in these regions has practical consequences on the future of agricultural production in the country, given that traditional agriculture has remained the mainstay of the CAR and other poor nations. Agriculture accounts for nearly half (49%) of CAR's GDP [67]. Ruan et al. [62] suggested that the expansion of the existing IoT infrastructure for agricultural applications would be necessary given that existing loads and connections cannot suffice. However,

the construction of IoT infrastructure for precision agriculture in open fields remains a challenge due to the low return on investment for private entities and the government. In the absence of sufficient economic incentives, it remains unclear whether investors would be willing to commit significant resources towards the deployment of IoT infrastructures [63]. The infrastructure-related challenges are further compounded by the need for accurate sensors with advanced capabilities and the urgency to resolve the disadvantages associated with various IoT protocols such as Zigbee, Bluetooth Low Energy (BLE), and Sigfox low-power wide-area network (LPWAN), among other networks. The concerns raised by Ruan et al. [62] and other scholars concerning the cost and unequal distribution of IoT infrastructure across the world can be countered by emerging benefits. Anthony et al. [37] claimed there were direct benefits associated with the deployment of IoT infrastructures in farms.

From the researcher's worldview, ensuring that existing IoT networks are compatible with other protocols, addressing signal interference, incompatibility with high power devices, and a lack of support infrastructure in certain countries would require significant financial resources [37]. Key stakeholders could be incentivized to support the expansion of the infrastructure through the demonstrated benefits. Previous studies had demonstrated that IoT infrastructure could yield cost savings of up to 500 USD/acre per crop cycle with the weather data-driven decision support systems and electrical capacitance sensors for predicting the optimal time for fungicide application, analysis of soil–water balance and soil–water content, demand-driven and intelligent irrigation of crops, and fertilizer application based on the plant's nitrogen needs [37]. The latter cost savings could help offset the technological barriers to IoT use in greenhouses.

The configuration and core systems embedded into IoT devices for greenhouses show that the transition from manual to autonomous systems is resource-intensive [35]; this makes it challenging for commercial farms to initiate the transition without a demonstrated proof of concept. The case for the resource-intensiveness of IoT and other systems in commercial agriculture contrast with Lara et al., who noted that it was practical to develop affordable/low-cost intelligent materials for greenhouse structures; this was partly achieved through the integration of advanced systems such as Wireless Sensor Network (WSN) embedded with 5G technology, WAN, or WiFi, for seamless collection and transmission of data. Cumulatively, an IoT prototype was successfully developed at the cost of 16 USD [29]; this is commendable considering that cost is a critical factor in agricultural production.

The observations made by Lara et al. [29] about the availability of low-cost sensors for greenhouses was corroborated by Placidi et al. [41], who noted that recent R&D efforts had facilitated the development of low-cost sensor devices capable of transducing the physical parameters and post-processing of raw data to obtain reliable information while complying with existing regulations; "the size and the cost of sensors have been reduced, making WSN the foremost driver of precision agriculture." Nonetheless, the purported reduction in the cost of the technology was application specific. The recent advances in R&D had not translated to the development of low-cost sensors for monitoring the soil water content. Accurate soil water content sensors vary between 150 USD and 5000 USD, according to Placidi et al. [41]; this exceeds the estimated average cost of a greenhouse in developing nations [27], and it is uneconomical to invest in expensive soil–water content sensors. The latter evidence demonstrates that the perceived costs reduction was sensor-specific; this challenge could be partly offset through research and the commercialization of existing innovations. The application-specific cost benefits of sensor technology were demonstrated in a joint report prepared by Cisco and the International Telecommunication Union (ITU) on harnessing IoT for global sustainable development [61]. The cost comparison data in Table 2 shows that sensors with advanced technical features such as WiFi connectivity were expensive (>150 USD/sensor).

**Table 2.** Cost comparison of different sensor types [49] (Reproduced with permission from publisher (Elsevier) from Castañeda-Miranda et al (2020)).

| Sensor Type | Approximate Cost in USD |
|---|---|
| Integrated WiFi High-Temperature Sensor | 204 |
| Integrated WiFi Humidity Sensor | 180 |
| Dual-Range Force Sensor | 110 |
| Ultrasonic Range Finder; $CO_2$ Sensor | 100 |
| Gas Pressure Sensor | 83 |
| Vernier Motion Detector | 75 |
| Oxygen Sensor | 60 |
| Wind Speed Sensor | 45 |
| Humidity & Temperature Sensor | 42 |
| Multichannel Gas Sensor; Liquid Level Sensor; Soil Moisture and Temperature; Liquid Level Sensor; GPS Breakout Sensor; Wearable GPS Module | 40 |
| G5 Water Flow Sensor; RFID Reader; RFID Sensor Module; Ultrasonic Range Finder Lite; Temperature and Humidity Sensor Board; | 30 |
| AC Current Sensor high amperage | 12 |
| Sound Detector | 11 |
| Capacitive Touch Sensor; Load Sensor (up to 50 kg); Air Quality Sensor; FM Receiver; Temperature and Humidity Probe; Barometric and Temperature Sensor; Altitude Sensor; Digital UV Index/IR/Visible Light Sensor; Proximity Light Sensor; Liquid Flow Meter; PIR Motion Sensor | 10 |
| Current Sensor; PIR Motion Sensor; Collision Sensor; Temperature and Humidity HP Sensor | 9 |
| RGP Light Sensor; Alcohol Sensor; RGB Color Sensor and IR Filter | 8 |
| Infrared shooting sensor; AC Current Sensor; HDR Digital Light Sensor; Microphone Amplifier | 7 |
| Luminosity Sensor; Loudness Sensor | 6 |
| Infrared reflective sensor; RFID Capsule; Ambient Light Sensor; Moisture Sensor; Sound Sensor | 5 |
| Vibration Sensor; Water Sensor | 3 |

Other fundamental issues of interest include the need to align corporate farming practices with global patterns to enhance commercial relevance. Leading industry stakeholders, including Mordor Intelligence (2021), postulated that the growth of the smart greenhouse market would be segmented into different categories (such as intelligent irrigation systems, valves and water pumps, sensors and cameras, control systems, HVAC, LED lights, and hydroponic/non-hydroponic technologies) and geographies (most growth would be recorded in advanced economies such as the US, Canada, UK, France, Germany, India, China, and Japan). Even though the segmentation of greenhouses had been poorly investigated in scholarly research, it provides compelling grounds for further research and the development of smart technologies for greenhouses. In other cases, the cost is a lesser barrier to the adoption of IoT in smart greenhouses relative to concerns about the utility of the technology. The reservations expressed by users of new technologies show that IoT-based systems must overcome social attitudes towards advanced technologies in agriculture. The technological-related concerns among end-users of the technologies could be linked to lesser demonstrated benefits.

Agrovoltaics and Blockchain

Agrovoltaics is also an emerging IoT-related technological revolution that has promising applications in sustainable energy-food production. Emerging developments include light transmittance and extended service life using foldable PV modules and solar tiles [16] and integrated electrical and thermal energy generation systems [68–70]. A key drawback is the nascency of agrovoltaic systems [71–73]. As of 2020, the only functional agrovoltaic system was in Belgium [73]. The newness of agrovoltaics is a limiting factor given the barriers to new technology adoption. The latest insights drawn from this review have practical significance in commercial agriculture. Additionally, further progress has been made in the commercialization of PV modules and solar tiles at the All-Russian Research

Institute of Electrification of Agriculture [69]. Beyond Russia and Belgium, it is anticipated that the demand for agrovoltaics would grow exponentially in line with the transition to renewable energy in agriculture. Beyond agrovoltaics, the integration of blockchain in IoT-based systems for farming is an emerging area that warrants further R&D attention [74,75]. Blockchain would help enhance security in agro-systems and decentralize solar energy generation and use.

## 4. Conclusions

The body of knowledge affirmed the practical benefits and limits of different IoT sensors for the optimization of the greenhouse microclimate. Current progress in research has demonstrated that it was feasible to remotely monitor the soil nutritional content, water/humidity and temperature, and real-time plant physiology (vegetative index, nutritional requirements, electrical conductivity, and magnetic susceptibility) using advanced sensors that feature Bragg, piezoelectric, electrochemical, electromagnetic, and fiber-optic technologies. Recent advances in technology have contributed to the development of advanced sensors made of graphene materials, multi-wavelength laser diode-photodiodes, rRNA biosensors, multiprobe chips, and microfluidic chips featuring nanocavity-enhanced photoelectrochemistry (Cu nanoparticles and the Cu-electrode). On the downside, certain challenges abound, including variable accuracy. On the one hand, there were highly reliable greenhouses sensors for commercial applications. Such sensors had translated to significant improvements in crop yields and cost savings through the data-driven decision support systems that provided alerts on when it was most appropriate to apply fertilizers, fungicides, and pesticides and irrigate crops that required less water for optimal growth.

The preliminary data drawn from published studies suggest that it was possible to achieve savings of up to 500 USD per acre cultivated. Considering that the costs were based on an isolated case study, the accrued savings could be higher or lower depending on the crops, type of sensors, and level of investment in data-driven decision support systems. On the other hand, there was a class of sensors with low accuracy of 2–25%, which was inadequate. The low accuracy of the sensors was a critical barrier to the commercialization of IoT sensors, and the problem was amplified by limited progress made in the development of ultrasensitive sensors and the unequal availability of IoT infrastructure in emerging and developed countries, and lack of sufficient incentives to facilitate investments in technology in rural areas. For example, internet connection in poor/lower-middle-income countries in Asia and Africa was below 25%, making it impractical for private investors to install IoT systems in smart greenhouses. The innovation-related challenges could be resolved through R&D. However, there was no immediate solution to the ICT infrastructural challenges in emerging nations. In light of the digital divide, the application of the internet of things for optimized green-house environments would remain a preserve of developed nations with advanced resources; this is evident from the investment in greenhouse technologies in Greece, Saudi Arabia, Europe, and North America. The unequal investments in smart greenhouses would have long-term implications on food security and agricultural sustainability in these regions, considering that global warming and climate change would result in inflation in the cost of agricultural produce. The inflation was attributed to the inflation in the cost of agricultural produce and unexpected severe weather events.

### 4.1. Limitations

IoT systems for the optimization of greenhouse microclimates are diverse. For example, sensors based on fiber-optic technologies or multi-wavelength laser diode-photodiode, RNA biosensors, microfluidic chips featuring nanocavity-enhanced photoelectrochemistry (Cu nanoparticles and the Cu-electrode), or graphene-based sensors offer distinct benefits and drawbacks compared to Bragg, piezoelectric, electrochemical, and electromagnetic sensors depending on the parameters to be measured, the influence of external variables, and the crops under cultivation. The current discourse did not review each class of sensor exhaustively, given that the primary focus was on the application of IoT for optimized

greenhouse environments and microclimates for crop production. The limitations were addressed in the next review article on the topic. The scope of the discussion was confined to the dominant themes in literature. The present discourse on precision agriculture is centered around cloud computing, IoT, and other technologies for precision agriculture [40], and commercially viable applications are either new, expensive, or poorly adapted to semi-arid conditions. The high cost of IoT application in the optimization of the greenhouse microclimate impended the commercialization of the technology by smallholder farmers.

### 4.2. Future Prospects

Even though there were varied challenges and barriers to the deployment of sensors and IoT systems and smart greenhouses, progress in R&D would translate to the widespread availability of low-cost and accurate sensors for water, soil nutrition, temperature, and humidity monitoring. The positive outlook is supported by the development of graphene (a material with high tensile strength, a light weight, flexibility, and ecological friendliness), high-precision rRNA biosensors, and microfluidic chips featuring nanocavity-enhanced photoelectrochemistry. However, it remains unclear whether future technologies would help to lower the cost of sensors with higher functionality (such as solution interoperability, industrial-scale deployment, long-term deployment, and accurate and precise measurements). At present, the inclusion of such capabilities in sensors cost 1,000 USD or more in a sensor [61]. Emerging technologies such as the LEO constellation would help provide global broadband internet coverage in remote errors, thus eliminating the capital and infrastructural barriers to mass IoT adoption in commercial agriculture.

**Author Contributions:** Manuscript preparation, C.M.; Review and edit, T.B. All authors have read and agreed to the published version of the manuscript.

**Funding:** This research received no external funding.

**Institutional Review Board Statement:** Not applicable.

**Informed Consent Statement:** Not applicable.

**Data Availability Statement:** Not applicable.

**Conflicts of Interest:** The authors declare no conflict of interest.

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
