# Peer review of "Application of Internet of Things (IoT) for Optimized Greenhouse Environments"

_agriengineering, doi:10.3390/agriengineering3040060_

Round 1

Reviewer 1 Report

The presented paper is an overview of modern technologies for effective monitoring and management of greenhouses. The topic is relevant and may be of interest to specialists and researchers in the fields of agriculture, solar energy, automation, since protected ground allows to create and manage the parameters of an environment for growing plants, when growing in open ground carries certain risks, which in recent years have become more and more are getting worse. The paper provides an overview of modern technologies for monitoring the main parameters of greenhouses and the authors have done a certain search and analysis of sources, however, as comments and recommendations, several points should be noted:

1. The overview of technologies in the introduction is broad, but their simple listing should be replaced with a brief description of the technologies themselves - their advantages and disadvantages.
2. On page 2 in line 69, source "[20]" follows before source "[19]" - authors should correct the sequence.
3. On page 3 in lines 101 "[25] - [25]" should be replaced with the correct sources.
4. On page 3 in lines 113 "[25] - [25]" should be replaced with the correct sources.
5. There is no reference to the source "[28]" in the text - the authors should indicate it.
6. There is no reference to the source "[55]" in the text - the authors should indicate it.
7. Authors should pay attention to the use of figures already from published papers, since they may have copyright protection and not abuse their use, but it is better to obtain permission to use them or remove them from the text and leave only a mention.
8. Also, the authors should pay attention to the compilation of tables - the content should not repeat the original published text, and the table should be described in their own words or significantly changed - add or remove information, otherwise such content may be regarded as plagiarism. In general, such tables should be significantly reduced or described in words the basic information recorded in them.
9. Is the optimum temperature for all plants and at all stages of growth 35 degrees Celsius and 95 % relative humidity? Authors should consider this point.
10. Information on page 13 from line 446 to line 466 should be removed or edited according to the work done.
11. Authors should slightly expand the review and add illustrations for a broader and in-depth overview of the topics under consideration. Moreover, illustrations should include the author's content, and not already published.
12. The contribution of the authors to the topic under consideration is also interesting, although it is clear that the paper is of an overview nature, but the proposals and conclusions of the authors themselves on the topic under consideration, their developments and research, etc. are interesting.
13. The authors should add a subsection in the paper on the use of Agrivoltaic systems, which are already being actively implemented all over the world in agriculture, including in protected ground, in greenhouses and can provide energy supply to both the sensors themselves and auxiliary equipment, pumps , lighting, etc.
14. In agrivoltaic systems, authors should pay attention to solar modules with light transmittance and extended service life (DOI: 10.4018/IJEOE.2020040106), and photovoltaic thermal modules, which can be performed both in planar concentrator design (DOI: 10.4018/978-1-5225-3867-7.ch004), thermal energy from which can also be used in working processes in greenhouses. Solar modules of this kind will find wide application in sunny and hot countries, in relation to which the study is underway in the paper.

In general, the presented paper leaves a positive impression, is of a review nature, and the sources used in the paper are quite fresh and published in reputable international journals. After eliminating the above remarks and taking into account the recommendations made, the presented paper can be published in the AgriEngineering journal.

Author Response

Please see our response in the attached file.

Reviewer 2 Report

This survey is in general well written. However, I found some parts not so attractive, as the authors are mostly reproducing conclusions and results from other papers, with very little new insights/opinions/suggestions. It would be much more interesting if the authors made a more critical analysis of the related work.

Moreover, some discussions are too shallow. For instance, let's take Table 2 as an example, the line about LoRa technology. It is said that Long-range (10 km) is an advantage, but that Actual line-of-sight range of ~2 km is a disadvantage. Obviously, there is a contradiction here. However, as the context of the table is not really discussed in detail in the paper, this is barely noticed. There are other inconsistencies in other technologies. Please remove this table, as its contents are imprecise or not relevant.

Finally, as the manuscript is on IoT applied to smart greenhouses, I miss a discussion related to the potentials and applications of blockchain technology. For instance, you may consider:

https://doi.org/10.1109/IOT-TUSCANY.2018.8373021

https://doi.org/10.1109/ACCESS.2020.2973178

but there are certainly many other related papers that could be used as references in this context.

Author Response

Please see our responses in the attached file

Round 2

Reviewer 1 Report

After corrections and additions made by the authors, the paper looks better and can be published in the journal AgriEngineering.  

Information on page 15 in lines 546 - 566 should be agreed by the authors with the Editorial board of the journal.

Reviewer 2 Report

I am satisfied with the changes made by the authors.